# Tracing the Evolutionary Expansion of a Hyperdiverse Antimicrobial Peptide Gene Family in *Mytilus* spp.: The MyticalinDB Resource

**DOI:** 10.3390/genes16070816

**Published:** 2025-07-12

**Authors:** Dona Kireta, Pietro Decarli, Damiano Riommi, Nicolò Gualandi, Samuele Greco, Alberto Pallavicini, Marco Gerdol

**Affiliations:** 1Department of Life Sciences, University of Trieste, 34127 Trieste, Italy; 2Department of Agronomy, Food, Natural Resources, Animals and Environment, University of Padova, 35020 Legnaro, Italy

**Keywords:** *Mytilus*, myticalins, antimicrobial peptides, AMPs, gills, expression, evolutionary expansion

## Abstract

**Background**: The overwhelming majority of the antimicrobial peptides (AMPs) studied in mussels (*Mytilus* spp.) so far are specifically expressed by hemocytes and display compact disulfide-stabilized structures. However, gill-specific myticalins play a role in mucosal immunity and are one of the very few examples of known molluscan AMPs lacking cysteine residues. **Methods**: We investigate the molecular evolution of myticalins, compiling a collection of sequences obtained by carefully annotating 169 genome assemblies of different *Mytilus* species. We determine the gene presence/absence patterns and gene expression profiles for the five myticalin subfamilies, including the newly reported myticalin E. **Results**: All sequences are deposited in MyticalinDB, a novel database that includes a total of 100 unique mature myticalin peptides encoded by 215 protein precursors, greatly enriching the compendium of these molecules from previous reports. Among the five subfamilies, myticalin A and C are the most widespread and highly expressed across all *Mytilus* species. Interestingly, structural prediction reveals a previously unreported strong amphipathic nature for some myticalins, which may be highly relevant for their biological activity. **Conclusions**: The results reported in this work support the role of myticalins in gill-associated mucosal immunity and highlight the importance of inter-individual molecular diversity in establishing an efficient response to microbial infections. The newly established MyticalinDB provides a valuable resource for investigating the evolution and extraordinary molecular diversity of this AMP family.

## 1. Introduction

Mussels, belonging to the genus *Mytilus* Linnaeus, 1758, are among the metazoans with the highest number of gene-encoded antimicrobial effectors described in the scientific literature [1]. Similar to other marine bivalves, these filter-feeding organisms are exposed to a plethora of different potentially pathogenic microbes. The abundance and diversity of these microbes undergo considerable short- and long-term changes in parallel with significant alterations in their habitat, characterized by tidal cycles, sudden changes in salinity, pH, dissolved oxygen, and nutrient availability [2]. In the absence of an adaptive immune system capable of providing a plastic response to novel antigens, bivalve mollusks have developed a sophisticated innate immune response involving a large and diverse repertoire of Pattern Recognition Receptors (PRRs), as well as an impressive arsenal of defense molecules [3,4].

In mussels, these were initially identified in hemocytes, the circulating cells that are believed to play a key role as the first line of defense against infection [5,6]. Studies conducted on defensins, mytilins, and myticins since the early 1990s represent one of the cornerstones of bivalve immunology [7,8,9]. These studies have also led to remarkable developments in more recent times, providing new interpretations regarding their accessory immunomodulatory and antiviral functions [10], their role in wound healing [11], and their extreme molecular diversity [12]. However, studies of AMPs from mussels and other bivalve mollusks have focused almost exclusively on peptides rich in cysteine residues. In these peptides, disulfide bridges are used to stabilize a complex compact structure, often based on a Cys-αβ motif, a feature shared by a multitude of natural defense peptides [13].

Comparatively, very little is known about other classes of AMPs that are widespread across taxa, or even predominant in certain major phyla, such as linear peptides characterized by strong compositional biases. For example, no known cases of glycine-rich peptides have been identified to date, either in mussels or any other bivalve species, despite their prominence as one of the major AMP types in arthropods [14,15]. Similarly, no bivalve AMPs adopting an amphipathic α-helical secondary structure have been described in bivalves, despite their widespread occurrence in diverse taxa [16].

Among the few known examples of AMPs devoid of cysteine residues in bivalves, CgPrps are particularly noteworthy. These short proline-rich peptides were identified in oyster hemocytes, where they appeared to act synergistically with defensins [17]. Molluscidins are another type of highly cationic linear AMPs that have, so far, only been identified in oysters and pen shells, and exhibit a remarkable sequence similarity to the namesake AMP found in abalones [18,19].

The only currently known example of linear AMPs in *Mytilus* is myticalins, which, like other mussel AMPs, are lineage-specific. Distantly related orthologous genes, named modiocalins, have, so far, only been identified in *Modiolus* spp. [20]. These peptides are produced as prepropeptides with a tripartite structure: an N-terminal signal peptide for secretion into the extracellular space, a highly cationic mature peptide in the central region, and a low-complexity, anionic C-terminal extension. The mature peptide and the C-terminal extension are separated by a dibasic site, presumably recognized and cleaved by a protease during protein maturation. Mature peptides exhibit a strong compositional bias, which allows for classification into four subfamilies—myticalins A, B, C, and D—based on the composition of amino acid residues. Although the lack of a complete genome for *Mytilus* only allowed for a partial overview of myticalin molecular diversity in the study by Leoni and colleagues [20], the little data available already indicated the presence of considerable inter-individual sequence variability in *Mytilus galloprovincialis*.

Myticalins display interesting antimicrobial properties, acting against a relatively broad range of Gram+ and Gram- bacteria, with different members having remarkably different target specificities [20]. Further studies have demonstrated that their antimicrobial action is exerted through membrane permeabilization [21], also exploring the effect of modifications aimed at increasing their potency and reducing their cytotoxicity in light of potential biotechnological applications [22,23].

One of the most intriguing features of myticalins is their tissue specificity, which localizes them to the gills, one of the largest contact interfaces with the external environment. Bivalve gills are covered by a mucosal layer and are essential not only for gas exchange, but also for filter feeding [24,25]. In bivalves, the gills represent a major potential entry point for pathogenic microorganisms and parasites [26], while also hosting microbiota that play a crucial physiological role in certain species [27,28,29]. Given the growing recognition of the importance of mucosal surfaces in the bivalve immune response [30], investigating the molecular components involved in their protection has become a key focus in current immunological research involving these organisms [31,32]. In this context, bivalve gills have previously been shown to be the preferential site of expression of several PRRs [33,34,35], which, in some cases, have been clearly implicated in a dual role in defense against pathogenic microbes and the regulation of beneficial symbionts [36]. In light of the strong gill-specific expression of myticalins, their remarkable sequence diversity, and their selective action towards certain bacteria, one might wonder whether these AMPs can contribute to a similar process in *Mytilus* spp.

Here, through an in-depth analysis of the available genomic resources for *Mytilus* spp., we report a comprehensive collection of myticalins. This collection comprises 215 different precursor protein sequences, corresponding to 100 unique mature peptides, including those belonging to the novel subfamily E. We discuss their evolution and molecular diversity in light of gene presence/absence variation (PAV) and provide a detailed estimate of their expression levels in the gills. All sequence data are made publicly available through MyticalinDB, a user-friendly online database that will facilitate future updates as new mussel genome assemblies become available.

## 2. Materials and Methods

### 2.1. Compilation of Mytilus Genomes

All publicly available genome assemblies and whole-genome resequencing datasets from six *Mytilus* species (*M. edulis*, *M. galloprovincialis*, *M. trossulus*, *M. californianus*, *M. chilensis*, and *M. coruscus*), available as of April 2025, were retrieved from the NCBI Datasets database, resulting in a dataset of 169 genomes (Appendix A).

For *M. galloprovincialis*, we included the de novo assembled genomes of 14 individuals [37]; MgalMED [38]; MytGallo_primary_0.1 [39]; PURA (ASM167691v1) [40]; LOLA v.mg3 [37]; and ASM4841453v1 (unpublished, NCBI WGS master JAQPZN0000000000.1). For *M. edulis*, we included MeduEUN and MeduEUS [39]; PEIMed_v2 [41]; xbMytEdul2.2 (unpublished, NCBI WGS master CAVMBL000000000); and MEDL1 [42]. Additionally, we incorporated one assembled genome from *M. californianus* xbMytCali1.0.p [43]; *M. trossulus* PNRI_Mtr1.2.1.hap1 (unpublished, NCBI WGS master JAZBVT000000000.1); *M. chilensis* [44]; *M. coruscus* MCOR1.1 [45]; Mcoruscus_HiC [46]; and SHE3 [47]. We also included the genome assembly mussel1.0 (unpublished, NCBI WGS master APJB000000000), described as *M. galloprovincialis*. However, due to the absence of pure *M. galloprovincialis* individuals in the sampling area (pers. comm. with Cynthia Riginos, September 2024), this genome most likely represents a hybrid of the native species *M. planulatus* and alien *M. galloprovincialis*.

Furthermore, 138 resequenced genomes from a recent study at the University of Exeter investigating global *Mytilus* population structure were also retrieved (unpublished, NCBI BioProject accession PRJNA932792). The sampling locations of individuals whose genomes were analyzed are reported in Appendix A.

### 2.2. Identification of Novel Myticalins

A set of previously published myticalin protein sequences [20] were individually BLASTed with tblastn against each of the databases listed in Appendix A using the CLC Genomics Workbench v25.0.1 (Qiagen, Hilden, Germany). The analysis was carried out with a threshold e-value of 10, a word size of 3, and the BLOSUM62 substitution matrix. Gap costs were set to existence and extension penalties of 11 and 1, respectively.

Putative novel myticalin genes were manually inspected and manually annotated based on a combination of Open Reading Frame prediction, the pairwise alignment between genomic DNA and cDNA sequences (whenever available), and the computational prediction of suitable splicing sites obtained with Genie [48]. The annotation process was focused on the coding sequence (CDS) only, included in exons 2 and 3 [20], whereas exon 1, which only comprises the 5′ untranslated region, was disregarded. Given the prevalence of pseudogenes, inclusion in the final dataset required verification that the CDS was complete and did not contain any loss-of-function mutations, such as nonsense or frameshift mutations and incorrect acceptor and donor splice sites. Nevertheless, fragmented gene sequences encoding mature peptides displaying 100% homology to known myticalins were considered evidence for the presence of a myticalin gene.

The CDS was virtually translated and the resulting amino acid sequences were manually annotated to identify the signal peptide, mature peptide, and C-terminal extension regions, as previously described [20]. Of note is that the mature peptide region is defined here as the peptide that would result from the proteolytic cleavage of the C-terminal extension before the occurrence of the additional post-translational modifications described in Section 3.3. Whenever a sequence lacked a canonical cleavage site, the C-terminal boundary of the mature peptide region was arbitrarily defined thanks to the multiple sequence alignment described in Section 2.3.

### 2.3. Phylogenetic Analysis

The complete amino acid sequences of all 215 myticalin precursors described in this study were included in a multiple sequence alignment (MSA) generated with MUSCLE v.5 [49] and manually refined to match the boundaries between exons and the signal peptide, mature peptide, and C-terminal extension regions. The resulting MSA was used as an input for a Maximum Likelihood phylogenetic inference analysis carried out with IQ-TREE v.3.0.0 [50]. ModelFinder [51] identified the HIVb+G4 model of molecular evolution as the best-fitting for the dataset according to the Bayesian Information Criterion. This model was implemented in the phylogenetic inference analysis, which was run with 1000 ultrafast bootstrap replicates. Due to the taxonomically restricted nature of myticalins, no feasible outgroups could be identified. Therefore, the phylogeny was presented as an unrooted tree with FigTree v.1.4.4 [52].

### 2.4. In Silico Gene Expression Analysis

As in the case of other hypervariable mussel immune effectors, obtaining an accurate quantification of the expression levels of single sequence variants belonging to the same group of evolutionarily closely related genes (in this case the A, B, C, D, and E subfamilies) is a challenging task. This is especially the case when the genotype of the analyzed individual is not known a priori. Therefore, we used a conservative approach that has been previously validated in other studies [53,54] to obtain a cumulative estimate of the expression levels of all members of the five myticalin subfamilies in the *Mytilus* species complex.

Briefly, we recovered publicly available RNA-seq datasets associated with different *M. galloprovincialis* tissues (i.e., hemocytes, mantle, and digestive gland), expanding this panel to the congeneric species *M. edulis*, *M. chilensis*, and *M. trossulus* for gills due to previous indications of strong tissue-specific expression [20]. Overall, the dataset included 44, 115, and 49 samples for the hemocyte, digestive gland, and mantle tissues, respectively. A total of 297 gill samples were analyzed. Following quality trimming carried out within the CLC genomics Workbench environment, reads were mapped against the reference genome LOLA v.mg3 [33,37] using stringent parameters (length fraction = 1 and similarity fraction = 0.75). The genome annotation was modified to replace associated myticalin genes with the annotated nucleotide coding sequences of the 215 myticalin precursors identified in this study. Gene read counts were then converted to Transcripts Per Million (TPM) [55] to ensure an appropriate comparison of expression levels across samples. The expression levels of myticalins were calculated as the cumulative sum of all the variants classified within each of the five subfamilies (A, B, C, D, and E). Tissue specificity was defined using the *tau* index [56]. The statistical significance of the differences in expression observed among tissues (i.e., gills vs. digestive gland, mantle, and hemocytes), as well as among the four major myticalins subfamilies (i.e., A, B, C, and D) in the gills, were assessed with unpaired and paired *t*-tests, respectively.

### 2.5. Three-Dimensional Structure Prediction

The amino acid sequences of the mature myticalin peptides were predicted with ColabFold v1.5.5 [57], leveraging the use of MMseqs2 [58] for the detection of suitable modeling templates and construction of sequence alignments and AlphaFold2 [59] for structural prediction. Peptides lacking the canonical dibasic propeptide cleavage site were excluded. Amino acid residues predicted to be cleaved during the post-translational proteolytic processing of the precursor protein (see Section 3.3) were removed prior to the prediction. Model confidence was assessed by inspecting the pLDDT scores. Specifically, we focused our attention on the models achieving the best scores [60] for each myticalin subfamily, which were visualized with ChimeraX v.1.9 [61].

## 3. Results and Discussion

### 3.1. MyticalinDB: An Updated Collection of Myticalin Sequences

From the comprehensive analysis of the 169 available *Mytilus* genome assemblies, we report the identification of a total of 100 unique mature myticalins, defined here as non-redundant peptides resulting from the processing of 215 different precursor proteins, explained in detail in Section 3.3. As the sequences of the mature peptides were predicted entirely in silico, it is important to note that their actual production in the form reported in this manuscript will require experimental confirmation in the future. Nevertheless, in line with previous publications [20,21], the amino acid sequences of the mature peptide were used as a reference for the nomenclature. In detail, each peptide was assigned a unique identifier, consisting of an uppercase letter indicating the myticalin subfamily (according to strict phylogenetic criteria; see Section 3.2) paired with a number. In cases where the same mature peptide resulted from multiple precursor proteins—differing by non-synonymous mutations in regions removed by proteolytic processing—a lowercase letter was added to the identifier to distinguish the corresponding protein precursor. Overall, 81 mature peptide sequences were associated with the genome assemblies of individuals belonging to the *Mytilus* species complex, and the remaining 19 peptides were found in the *M. californinanus*/*M. coruscus* lineage. Note that although several distinct species are recognized within the *Mytilus* species complex (*M. galloprovincialis*, *M. edulis*, *M. trossulus*, and *M. chilensis* have available genome assemblies), due to interfertility among them, extensive hybrid zones exist—some of which have formed recently—as well as well-documented genetic introgression [62,63,64]. As a result, the same mature myticalin peptides may be found in individuals from different species and/or geographical regions. It is possible and likely that differences in species identity and sampling location could have an effect on the pattern of myticalin sequences recovered, as is already apparent for some subclasses (e.g., B and D, discussed below). While our dataset includes genome assemblies from a wide range of *Mytilus* species and global locations, it remains incomplete—particularly for Southern Hemisphere species and the *M. californianus*/*M. coruscus* lineage, which is represented by comparatively few individuals. As more geographically diverse genomic data become available, MyticalinDB will provide an ideal tool to track and analyze potential spatial patterns in myticalin gene presence and expression.

The identification of 100 distinct mature myticalin peptides is a major advance compared with our previous report of only 27 sequences, most of which were not associated with a full-length precursor [20]. Notably, four of the sequences reported in the previous study (A1, A2, A6, and D1) were not detected in any of the genome assemblies analyzed in this study. Four additional sequences (A7, A10, D6, and C5) were only detected as partial sequences or as sequences characterized by the presence of nonsense mutations in at least one genome, suggesting that they could be pseudogenes. Due to uncertainties concerning their functionality, these eight sequences were not included in MyticalinDB and were omitted from further consideration in this study.

With the aim to provide user-friendly access to the myticalin sequences reported in this study, as well as those that may be discovered in the coming years, we developed MyticalinDB, a new online database accessible at https://myticalindb.units.it/. The main entries in MyticalinDB are mature peptide sequences, however, each mature peptide is linked with different associated precursor variants, allowing for the inspection and download of nucleotide and amino acid sequences for each entry in a FASTA format. Besides full-length sequences, MyticalinDB allows for the recovery of the signal peptide, mature peptide, and C-terminal extension regions individually. In addition, users can browse, search, and download data related to individual peptides or peptide classes. The database also supports the visualization of multiple sequence alignments, both for full-length sequences and for specific regions, enabling detailed comparative analyses of molecular diversity. Importantly, each sequence deposited in MyticalinDB is placed within an appropriate phylogenetic and evolutionary framework (discussed in detail in Section 3.2). The visualization of phylogenetic trees allows users to explore relationships among variants, and the association of each variant with individual genome assemblies enables gathering information on their distribution across different geographic sampling locations, providing a comprehensive overview of gene PAV.

MyticalinDB is designed as a dynamic and expandable platform: it supports continuous updates with newly identified variants and allows for the reassociation of previously known sequences to additional genomes as they become available. This will most certainly be required in the future; due to the high fragmentation of most of the genome assemblies analyzed in this study, sequenced with low coverage and using short reads only, a high number of partial gene sequences were identified that, in some cases, may correspond to novel myticalins which were not included in the present collection of sequences. Moreover, MyticalinDB offers the potential to incorporate related sequences from other *Mytilus* species and even evolutionarily related AMPs that may be discovered in non-*Mytilus* taxa in the future, similar to that observed with the case of modiocalins, present in the *Modiolus* genus [21]. As such, MyticalinDB represents a robust and flexible tool for the study of myticalin sequence diversity and evolution.

### 3.2. Molecular Evolution of Myticalins

The updated phylogenetic inference analysis of all the myticalin precursor protein sequences recovered in this study confirmed the presence of the four distinct previously described subfamilies (A, B, C, and D) [20] and revealed the presence of the additional subfamily E (Figure 1). The monophyly of the five myticalin subfamilies was strongly supported by high bootstrap values (i.e., 100 in all cases) and was consistent with the main structural features of the mature peptides and their amino acid compositions, as described in Section 3.3. Despite substantial divergence at the primary sequence level, the five myticalin subfamilies most likely originated from a single ancestral precursor gene. As we have previously demonstrated [20], all myticalin genes exhibit a conserved exon–intron structure, comprising three exons and two introns; the second intron is consistently a phase one intron, interrupting the coding sequence at a conserved position. Notably, this architectural feature was also retained in the newly identified myticalin E subfamily. Furthermore, in all chromosome-scale genome assemblies of *Mytilus*, myticalin genes from different subfamilies were found clustered within a few kilobases of each other, providing strong evidence for their emergence via tandem duplication events followed by the subfamily-specific divergence of paralogous genes (Appendix A).

The identification of at least one member of four myticalin subfamilies (i.e., A, B, C, and E) in both the *Mytilus* species complex and in the *M. californianus*/*M. coruscus* lineage (Figure 1) strongly supports an ancestral diversification of these molecules, predating the split between these two evolutionary lineages, estimated to have occurred approximately 78 million years ago [37]. Although no coding myticalin D gene was identified in either *M. californianus* or *M. coruscus*, the occasional presence of D subfamily pseudogenes is also consistent with the presence of the five subfamilies in the ancestral *Mytilus* species. Gene PAV and pseudogenization events may have subsequently led to lineage-specific expansion, shrinkage, or a near-complete loss of some myticalin subfamilies in the two divergent mussel lineages, as detailed in Section 3.3.

Besides the clear identification of the five myticalin subfamilies reported above, phylogenetic inference also allowed for the identification of significant molecular differentiation events within subfamilies A and D. In the former, the sequences from the *M. californianus*/*M. coruscus* lineage were grouped in a divergent, highly supported clade (bootstrap = 100, Figure 1), whose peculiar structural features are discussed in detail in Section 3.3.1. In the latter, two distinct, well-supported groups of sequences were also clearly identified in the tree, easily recognized by differences in amino acid composition—discussed in Section 3.3.4.

### 3.3. Myticalin Sequence Features

As previously reported [20], all myticalin precursors are characterized by a tripartite organization consisting of an N-terminal signal peptide, a mature peptide—which consistently retains a strongly cationic nature despite subfamily-specific length and amino acid variability—and an anionic C-terminal extension.

During maturation, myticalins are predicted to undergo two consecutive proteolytic cleavages. The first, carried out by signal peptidase, occurs within the endoplasmic reticulum immediately after the hydrophobic core of the signal peptide. This takes place once the protein/ribosome complex is directed to the translocon by the signal recognition particle, as is typical of canonical secretion [65]. The second cleavage is predicted to occur later at the C-terminal end of the mature peptide, immediately after a dibasic site (Lys-Arg or, less frequently, Arg-Arg), which is found in a highly conserved position in nearly all myticalin precursors. Although this remains to be experimentally validated, this cleavage is most likely mediated by a proprotein convertase whose recognition site is present in other known mussel immune effector precursors, such as big defensins and CRP-I [54,66]. As explained in the next sections, this conserved site occasionally displays nonsynonymous mutations, which may, therefore, indicate the pseudogenic nature of some myticalin sequences. However, since we cannot presently rule out the possibility that these unconventional sites may still undergo proteolytic cleavage by alternative means, these precursor sequences have been included in the dataset reported in the present manuscript, even though they are marked with a warning flag in MyticalinDB.

The possibility that other post-translational modifications may target myticalins is uncertain due to the lack of experimental evidence. This is mainly due to the difficulty of isolating enough material for biochemical analysis, since these peptides are only expressed in gills and vary between individuals, necessitating both pooling and genotyping (to identify individuals carrying the same variants), which cannot be performed on the same individuals. Nevertheless, a significant portion of myticalin peptides contain a conserved glycine residue preceding the dibasic cleavage site, which has been previously interpreted as an amidation site [20,21]. This modification, previously documented in a broad range of other AMPs [67,68,69], would be enabled by the enzymatic action of peptidylglycine α-amidating monooxygenase (PAM) after the removal of C-terminal basic residues by carboxypeptidase B (CPB)-like enzymes.

Below, we will describe the main structural features of the five myticalin subfamilies in light of these general characteristics.

#### 3.3.1. A Subfamily

We identified a total of 25 myticalin A mature peptides (corresponding to 74 precursors) compared to the ten sequences described in the study by Leoni and colleagues [20]. Of these, 17 peptides were found in the *Mytilus* species complex and 8 were found in the *M. californianus*/*M. coruscus* lineage. As previously mentioned, no complete gene corresponding to five of the eight subfamily A peptides from the original study (i.e., A1, A2, A6, A7, and A10) could be identified. All mature peptides in this subfamily were strongly enriched in proline, arginine, and tyrosine residues (accounting for, on average, 27%, 25%, and 16% amino acid content, respectively). This was consistent with the presence of a variable number of PRX repeats (Figure 2A), where the third amino acid of the triplet was most often tyrosine or another residue with a hydrophobic side chain (e.g., Trp, Phe, Ile, Leu, or Met). The mature peptides ranged in size from 26 to 37 amino acids (aa). With the lone exception of myticalin A13, the peptides invariably contained a potential C-terminal amidation signal, likely targeted by PAM upon the removal of the three C-terminal basic residues (Figure 2B).

Although the sequences of the mature peptides of myticalin A in *M. californianus* and *M. coruscus* were very similar to those of the other mussel species, their precursor proteins were easily distinguishable. They contained a notably shorter C-terminal extension, comprising only 37-51 aa compared to >70 residues in variants from the *Mytilus* species complex (Figure 2A). These discrepancies also determined the placement of the myticalin A sequences from *M. californianus*/*M. coruscus* in a distinctly divergent branch of the phylogenetic tree (Figure 1, starred branches). Of note, the partial deletion or premature truncation of the C-terminal extension was also occasionally observed in a few *Mytilus* species complex precursors (i.e., A3g, A4b, A8b, and A8i, for details see https://myticalindb.units.it/), but the functional implications of these sequence variants are unknown.

Myticalin A genes were identified in nearly all analyzed *Mytilus* species complex genomes (i.e., 159 out of 165, 96% of the total), and the few missing occurrences most likely derive from highly fragmented low-coverage genome assemblies. Similarly, myticalin A genes were found in all analyzed *M. californianus* and *M. coruscus* genomes. The observation of an average of 1.75 myticalin A sequences per genome, with a maximum observed number of 6 sequences in the PM1 individual, strongly supports the presence of multiple paralogous myticalin A genes. Myticalin A8 was the most widespread variant in the *Mytilus* species complex (62 out of 165 individuals), followed by myticalin A5 (53 individuals) and A4 (41 individuals).

On rare occasions, a few myticalin A genes were detected as pseudogenic alleles, characterized by the presence of truncated or entirely deleted exons and frameshift or nonsense mutations (Figure 2C). Moreover, although their distribution was not tracked in detail, highly degenerated myticalin A paralogous pseudogenes were often observed, altogether mirroring the situation previously described for other highly diversified mussel immune effectors undergoing fast molecular evolution [53,70].

#### 3.3.2. B Subfamily

We identified nine mature myticalin B peptides (corresponding to 13 distinct protein precursors), significantly expanding the size of this subfamily compared to the previous study [20], where only a single peptide (B1) was described. Four of these sequences were associated with the *Mytilus* species complex, whereas the other four were found in the *M. californianus/M. coruscus* lineage. As seen in Figure 3A,B, despite significant length variations (from 17 to 38 aa), the mature peptides were predicted in nearly all cases to undergo the proteolytic removal of three C-terminal basic residues by CPB followed by C-terminal amidation by PAM, as was also the case for subfamily A. The mature peptides were strongly enriched in arginine, proline, and serine residues (accounting for, on average, 27%, 16%, and 16%, respectively). However, unlike in A myticalins, in this case, the Pro and Arg enrichment did not result in PRX repeats. Myticalin B precursor sequences also showed remarkable diversity in the length of the C-terminal extension region, which ranged from 67 to 91 aa, due to a variable number of repeats of an 8 aa-long motif (Figure 3A).

The presence of potentially functional genes associated with subfamily B was quite rare within the *Mytilus* species complex, with an average of 0.18 genes per individual. Only sporadic occurrences of multiple paralogous gene copies were found, even though degenerated B-type pseudogenes were observed in a higher fraction of individuals. This pattern highlights the high prevalence of PAV, which affected 135 out of the 165 analyzed genomes (82% of the total). Out of the four peptide variants identified in the *Mytilus* species complex, myticalin B2 was the most frequently observed (16 genomes), followed by B3 (8 genomes) and B1 (4 genomes) (Figure 3C). Curiously, these four peptides were encoded by precursor proteins that were uniform in length and exhibited a low sequence diversity, in stark contrast with the more diverse precursor sequences found in *M. californianus* and *M. coruscus*. The occasional detection of myticalin B2 and B3 only as pseudogenic alleles raises further questions about the biological relevance of myticalin B in the *Mytilus* species complex. In contrast, up to three different paralogous myticalin B genes were observed per genome in *M. californianus* and *M. coruscus*, suggesting a more widespread presence of this subfamily in these species compared with the *Mytilus* complex, although the small number of genomes available prevents a comprehensive assessment.

#### 3.3.3. C Subfamily

Myticalin C was the largest myticalin subfamily, with 36 mature peptides and 94 precursor sequences. Of these, 31 were found in species belonging to the *Mytilus* complex, and only 5 were found in the *M. californianus*/*M. coruscus* lineage. Of the ten myticalin C sequences described in the original report by Leoni and colleagues [20], only one (i.e., C5) could not be confirmed at the genomic level in this study. Despite their considerable diversity, myticalin C peptides were invariably strongly enriched in arginine (31% content, on average), but lacked proline residues, marking a striking difference compared with the four other myticalin subfamilies. These arginine residues were typically organized in consecutive stretches at the N-terminal end of the mature peptide, which ranged between 17 and 33 aa in length. In contrast, the C-terminal region showed a shift in amino acid composition, where acidic residues (Asp or Glu) were often present, marking another peculiarity of the myticalin C subfamily (Figure 4A). Compared with the other myticalin subfamilies, myticalin C precursors displayed a low size variability in their C-terminal extensions, which ranged between 45 and 54 aa, and unlike myticalins A and B, no C precursor displayed evidence of potential amidation (Figure 4B). Although a few precursors could undergo additional proteolytic cleavage by CPB following proprotein convertase processing, this process would not result in free C-terminal glycine residues. On the other hand, it is worth mentioning that most processed myticalin C peptides would display a conserved C-terminal aspartic acid residue, whose functional role is presently unknown. The precursor sequences found in the *M. californianus*/*M. coruscus* lineage did not display any recognizable feature that would allow for immediate visual discrimination from those of the *Mytilus* species complex, which is consistent with their intermixed placement in the phylogenetic tree (Figure 1).

As with myticalin A, myticalin C genes were identified in the overwhelming majority of *Mytilus* species complex genomes analyzed (i.e., 159 out of 165—96% of the total). The apparent lack of sequences in the remaining six individuals was most likely due to the high fragmentation of genomes from the low sequencing coverage. Complete myticalin C genes were also detected in three out of the four available *M. californianus*/*M. coruscus* genome assemblies. The identification of an average of 2.12 sequences per genome, with a maximum number of 9 different sequences in the PM1 individual, indicates that multiple paralogous gene copies might have been present. The most common myticalin C peptides in the *Mytilus* species complex were C3 (present in 58 out of 165 individuals), C2 (24 individuals), and C1 (21 individuals).

It is worth mentioning that a few of the myticalin C peptides we reported may be not fully functional and could be re-categorized as pseudogenes in the future. For example, myticalins C7a and C23 display non-synonymous mutations that would disrupt the propeptide dibasic cleavage site, even though both sequences still display a dibasic site potentially available for cleavage in a slightly offset position (Figure 4A). Other sequences (C20, C22, and C31) often detected as pseudogenic alleles (Figure 4C) or lacking evidence of expression (C21, C30, and C33, see Section 3.4) were also questionable. As in the case of myticalin A, the detection of additional highly degenerated myticalin C pseudogenes was not an infrequent event, indicating that some of the myticalin C sequences reported in this study may be undergoing the first stages of pseudogenization.

#### 3.3.4. D Subfamily

Our genomic investigations led to the identification of 27 different myticalin D mature peptides, corresponding to 30 distinct precursor proteins. All these sequences were identified in species belonging to the *Mytilus* complex, whereas only degenerated pseudogenes of this subfamily were occasionally detected in *M. californianus* and *M. coruscus*. Two of the seven sequences reported in the original publication (i.e., myticalin D1 and D6) [20] could not be validated at the genome level in this study. Despite their close phylogenetic relatedness (see Section 3.2), myticalin D peptides were divided into two subtypes based on their amino acid composition and mature peptide sequences. In detail, as clearly visible in Figure 5A, the first myticalin D subtype had high average contents of proline (27%), Arg (13%), and Thr (15%). On the other hand, the second subtype—despite maintaining high enrichment in both proline (24%) and arginine (27%)—also had a very high content of tryptophan residues (12%). Myticalin D mature peptides varied considerably in size from 27 to 47 aa. Like myticalin C, they displayed a marked shift in amino acid content towards their C-terminal end, where a highly conserved SATI(N/D/G)T(D/E) motif, shared by both subtypes, was visible (Figure 5A). Another point of similarity with myticalin C was the lack of detectable C-terminal amidation signals, as the proteolytic cleavage by proprotein convertase, followed by the removal of C-terminal basic residues by CPB, would most often produce peptides with free C-terminal Gln or His residues (Figure 5B). Although the C-terminal extension of the precursors encoding the Thr-enriched peptides was sometimes slightly shorter than those encoding Trp-enriched peptides (42–45 vs. 50 aa), their overall organization was very similar.

The distribution pattern of myticalin D genes in the *Mytilus* species complex was strongly suggestive of widespread PAV. Indeed, potentially functional myticalin D genes were only found in 103 out of 165 genome assemblies (i.e., 63% of the total), even though highly degenerated pseudogenes were observed in a few additional individuals. Although the average number of sequences detected per genome was rather low (i.e., 1.12) compared to myticalins A and C, the detection of a high number of sequences in some genomes (i.e., up to seven in PM1) indicates the possibility that multiple paralogous genes were present in the same haplotype.

Notably, some myticalin D precursor sequences have features that put into question their biological functionality. In particular, myticalins D3, D9, and D14, which all belong to the Thr-enriched subtype, display a missense mutation predicted to detrimentally affect the propeptide dibasic cleavage site. Moreover, other sequences (i.e., D7, D8, and D12) were often observed as pseudogenic variants (Figure 5C) or lacked evidence of expression (D21, D24, and D26; see Section 3.4) and were, therefore, flagged as potentially non-functional.

#### 3.3.5. E Subfamily

The myticalin E subfamily represents a novel group within the myticalin gene family. We identified only three complete mature peptides corresponding to four precursor proteins, two of which were found in the *M. californianus*/*M. coruscus* lineage. The peptides exhibited an amino acid composition enriched in tryptophan (W), proline (P), threonine (T), and arginine (R) and ranged in size from 37 to 39 amino acids, while the C-terminal extension ranged from 39 to 51 amino acids (Figure 6A).

Myticalin E genes were located within the same gene clusters of the other myticalin subfamilies (Appendix A), strongly suggesting their shared ancestral origin via tandem duplications. However, the lone detection of a myticalin E sequence in a single *M. edulis* complex genome (GA1) suggests that this subfamily may have been subsequently lost in the overwhelming majority of individuals belonging to this lineage (Figure 6C). This is further supported by the implied loss of proper maturation and, potentially, function. The peptides lacked a canonical proprotein convertase recognition motif in the expected position, which is typically required for post-translational processing, even though a dibasic site potentially available for cleavage would still be present in a slightly offset position in myticalin E1 and E2 (Figure 6A,B). Moreover, the absence of any expression data for myticalin E peptides (see Section 3.4) raises further doubts about their biological relevance. Taken together, these observations suggest that myticalin E may represent an evolutionary relic—a remnant of once-functional genes that are now degenerating or already non-functional. Future studies, including transcriptomic validation and antimicrobial testing, will be essential to determine whether this subfamily retains any biological activity or is indeed a vestigial remnant. While only a limited number of genomes are currently available for *M. coruscus* and *M. californianus*, ongoing genome sequencing efforts in these and other *Mytilinae* subspecies may eventually reveal whether myticalin E-related genes are present in extant mussel species evolutionarily related to *Mytilus* spp.

### 3.4. Overview of Myticalin Expression Levels

The marked gill-specificity of myticalins was preliminarily characterized in our previous study [20]. However, assessing the expression levels of myticalin transcripts with traditional targeted approaches, such as qRT-PCR, remains challenging. The difficulty arises from gene PAV and sequence hypervariability, which can lead to either a lack of amplification of certain target myticalins or the nonspecific amplification of similar paralogous genes due to primer mispairing.

Here, we used an in silico approach, similar to the one we previously applied to other hypervariable mussel gene families [54], to accurately estimate the expression levels of myticalins in available RNA-seq datasets from multiple tissues. Although this approach cannot always identify the specific myticalin variants expressed in each sample due to the possibility of read cross-mapping among highly similar sequences, it can provide an accurate estimate of the cumulative expression level of all members across the five myticalin subfamilies.

Our data confirmed the strong gill-specific expression of myticalins. The average observed cumulative expression level in this tissue was significantly higher (*p* < 0.05) than all other tissues. In detail, it was >100× higher than the mantle, >250× higher than the digestive gland, and >5000× higher than the hemocytes, with a calculated *tau* value higher than 0.99 (Figure 7E). This pattern applied to all myticalin subfamilies, with A, C, and D genes displaying significantly higher expression levels in the gills compared with the other three tissues and *tau* values equal to one (Figure 7A,C,D). Although this marked tissue specificity was not supported by statistical significance for B myticalins due to widespread PAV, these sequences were expressed at values of >1 TPM in 12.4% of gill samples and less than 1% of samples linked with the other tissues, suggesting a preserved preferential expression (Figure 7B). Consistent with their extremely low prevalence within the *Mytilus* species complex and possible pseudogenic nature, no evidence of expression was observed for myticalins belonging to subfamily E.

We demonstrate here that the cumulative expression level of myticalins was far from negligible, which is in line with our expectation that gene PAV may lead to significant underestimations of the biological relevance of myticalins measured with qRT-PCR. Mean and median expression values reached 3420 and 2166 TPM, respectively, with maximum peaks close to 30,000 TPM. Of the four subfamilies, myticalin C had the most consistent expression (mean = 1705 TPM, median = 1250 TPM, expression detected in 99.3% gill samples), followed by myticalin A (mean = 1076 TPM, median = 141 TPM, expression detected in 92.7% gill samples) and myticalin D (mean = 611 TPM, median = 23 TPM, expression detected in 70.1% gill samples) (Figure 7F). These differences were supported, in all cases, by statistically significant *p*-values (i.e., 0.05). However, in light of our genomic survey (see Section 3.3.4), the apparently lower expression levels of myticalin D were likely due to a higher occurrence of PAV and, therefore, they should not be interpreted as an indication of a lower biological relevance. Similarly, the lower expression level of myticalin B (i.e., 28 TPM) was consistent with the rare observation of these variants in resequenced individuals, only being detected in 17.9% of the analyzed gill samples. Future studies incorporating long-read RNA-seq could allow for more precise characterizations of individual myticalin transcript diversity and cellular expression patterns. In particular, it could clarify whether there are differences in the cell-type specific expressions of individual variants, which cannot be fully resolved using bulk RNA-seq.

### 3.5. Considerations About the Structure of Myticalins

In their original description, myticalins were classified as “linear” AMPs due to the absence of cysteine residues. This feature distinguishes them from most other AMP families previously identified in bivalves, whose 3D structure is stabilized by multiple disulfide bonds. Moreover, circular dichroism spectra analyses have further suggested that myticalins may lack definitive α-helical or β-sheet conformations [21,22]. However, since the purification of native myticalins in sufficient quantities for crystallographic structural determination remains challenging, these data were collected through the analysis of a limited number of synthetic peptides. We report here a preliminary exploration of the structural properties of myticalins, thanks to the use of AlphaFold2 for predictive modeling [59]. Although the quality of the models was generally not satisfactory for the members of the myticalin B and D subfamilies, meaningful results were obtained for myticalins A and C. In detail, the presence of PRY/PRW repeats in myticalin A mature peptides led to the prediction of a backbone characterized by a periodic torsion, with about three residues per turn, which would exactly match the characteristics of polyproline II helices [71] (Figure 8A). This organization would result in a spatial segregation of charged arginine and aromatic (tyrosine/tryptophan) residues within the PRY/PRW repeats, generating amphipathicity, a feature critical for membrane disruption in AMPs. On the other hand, subfamily C peptides yielded high-confidence predictions (pLDDT > 80) that supported, in nearly all cases, α-helical conformations. The models often supported a characteristic kink between the arginine-rich N-terminal region and the C-terminal segment enriched in asparagine and threonine residues (Figure 8B,C).

These predictions are inherently characterized by a high degree of uncertainty, partly due to our limited understanding of the potential additional post-translational modifications that may affect myticalins and partly due to the possibility that seemingly disordered peptides in aqueous solution may adopt ordered structures in hydrophobic environments, an occurrence documented for other invertebrate AMPs with strong compositional biases [72,73]. Notably, unlike myticalins A and B, the predicted structures of myticalins C and D remain largely unresolved, thereby limiting our ability to predict their interactions with the biological membranes of target microorganisms.

## 4. Conclusions

By coupling genomic, transcriptomic, and structural analyses, this study presents the most comprehensive survey of the myticalin antimicrobial peptide family in mussels to date, uncovering a remarkable diversity of sequences and expanding the classification to five distinct subfamilies, including the newly identified subfamily E. Although the reported mature peptide sequences, deriving from bioinformatic predictions, will undoubtedly require a rigorous experimental validation, our results emphasize the tissue-specific expression of myticalins in gills, their strong compositional biases, and their potential for functional specialization based on sequence variation and structure.

Notably, the prevalence of gene PAV across the *Mytilus* species complex underscores the dynamic nature of the mussel immune gene repertoire, mirroring previous reports on other AMP families from mussels [12,53,54,66] and other bivalves [74]. This high degree of PAV, combined with extensive structural variation, points to a complex evolutionary mechanism underlying immune diversification, likely driven by local adaptation to distinct microbial communities. Comparable evolutionary dynamics have been observed in the genomes of other species encoding a large number of gene products with exogenous targets, collectively referred to as the “exogenome” [75]. Although only a few such cases have been documented to date, the most striking example is arguably found in cone snails, where venom peptides have evolved under the selective pressure of intricate prey–predator interactions, leading to extreme inter and intraspecific sequence diversity [76]. While the evolutionary landscape of the mussel–pathogen interaction clearly differs from the predator–prey arms race observed in cone snails, mussel AMPs (as evidenced in myticalins) and conopeptides share remarkable similarities. These include an exceptionally rapid rate of evolution in their mature peptide regions, compared not only to other segments of the precursor protein, but also to gene products with endogenous targets. Together, these parallels suggest that convergent evolutionary strategies may arise across vastly different biological systems when selective pressures consistently act on genes mediating external biological interactions, underscoring the exogenome as a potential hotspot of adaptive innovation.

By making all sequences discovered here publicly accessible through MyticalinDB, a dedicated online website, our work provides a valuable resource for future investigations into the evolution, function, and biotechnological potential of these mussel immune effectors.

## Figures and Tables

**Figure 1 genes-16-00816-f001:**
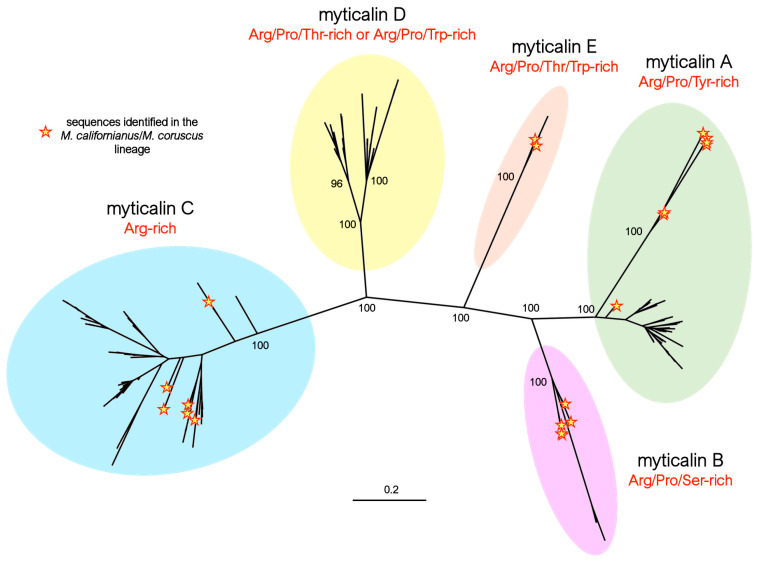
Maximum likelihood phylogeny of myticalins, highlighting the distinction among the five myticalin subfamilies (A, B, C, D, and E). For the sake of simplicity, only the bootstrap support values of the major nodes of the tree are shown. The placement of the myticalin sequences identified in the *Mytilus californianus*/*M. coruscus* lineage are highlighted by yellow stars.

**Figure 2 genes-16-00816-f002:**
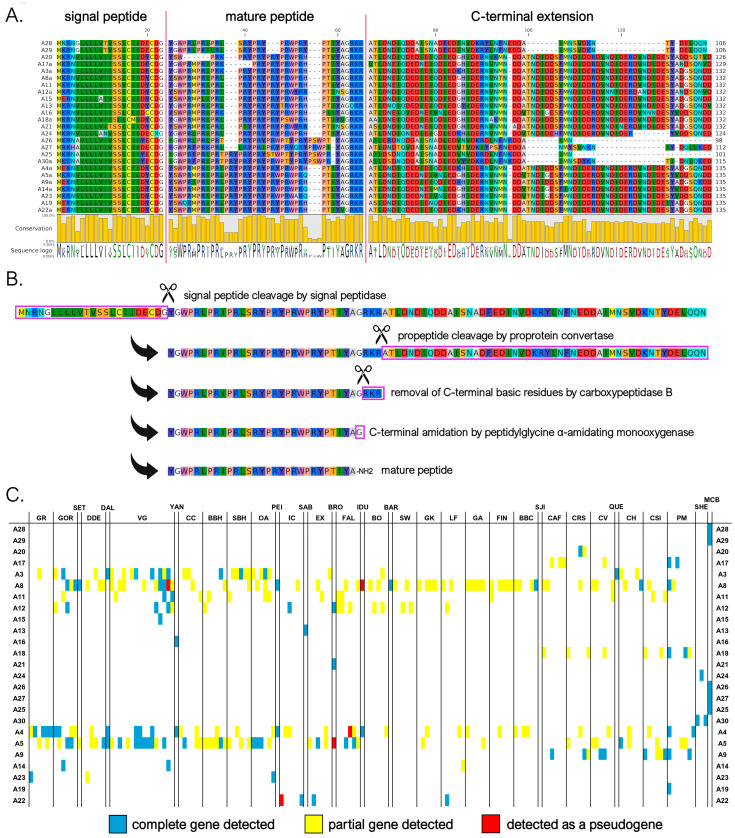
(**A**) Class A myticalins displayed in a multiple sequence alignment; (**B**) the events predicted to occur during the maturation of the precursor proteins, exemplified by myticalin A28; and (**C**) the occurrence of the sequences in sampling locations, where colors denote whether a complete (blue), partial (yellow), or pseudo (red) gene were detected. Please note that only a single representative precursor sequence is shown for each mature peptide. The complete multiple sequence alignment including all precursor sequences is available online on MyticalinDB. Note: C-terminal amidation does not occur in myticalin A13.

**Figure 3 genes-16-00816-f003:**
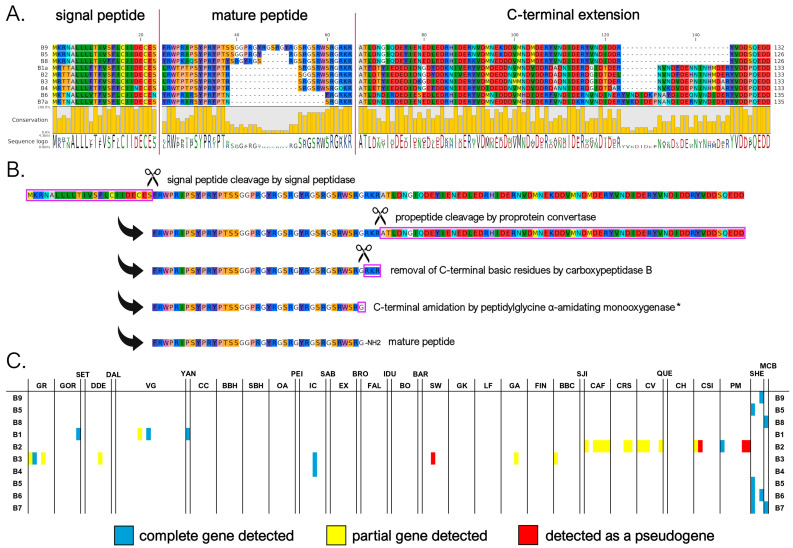
(**A**) Class B myticalins displayed in a multiple sequence alignment; (**B**) the events predicted to occur during the maturation of the precursor proteins, exemplified by myticalin B9; and (**C**) the occurrence of the sequences in sampling locations, where colors denote whether a complete (blue), partial (yellow), or pseudo (red) gene were detected. Please note that only a single representative precursor sequence is shown for each mature peptide. The complete multiple sequence alignment including all precursor sequences is available online on MyticalinDB. * C-terminal amidation does not occur in myticalin B4.

**Figure 4 genes-16-00816-f004:**
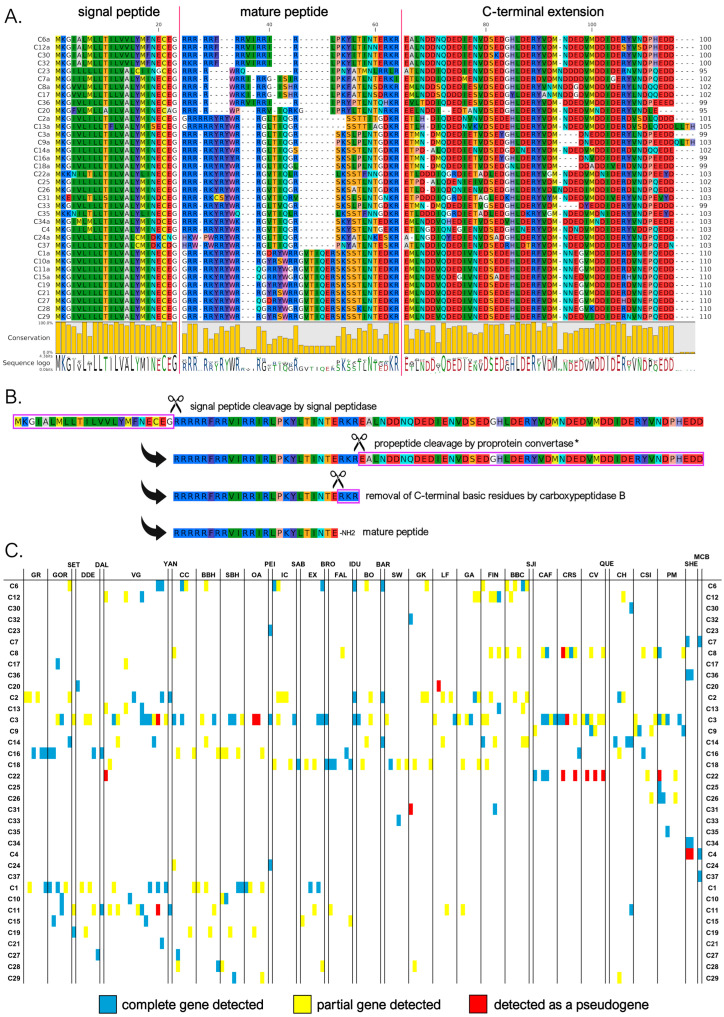
(**A**) Class C myticalins displayed in a multiple sequence alignment; (**B**) the events predicted to occur during the maturation of the precursor proteins, exemplified by myticalin C6a; and (**C**) the occurrence of the sequences in sampling locations, where colors denote whether a complete (blue), partial (yellow), or pseudo (red) gene were detected. Please note that only a single representative precursor sequence is shown for each mature peptide. The complete multiple sequence alignment including all precursor sequences is available online on MyticalinDB. * Note the disruption of the canonical dibasic site in myticalin C7a and C23.

**Figure 5 genes-16-00816-f005:**
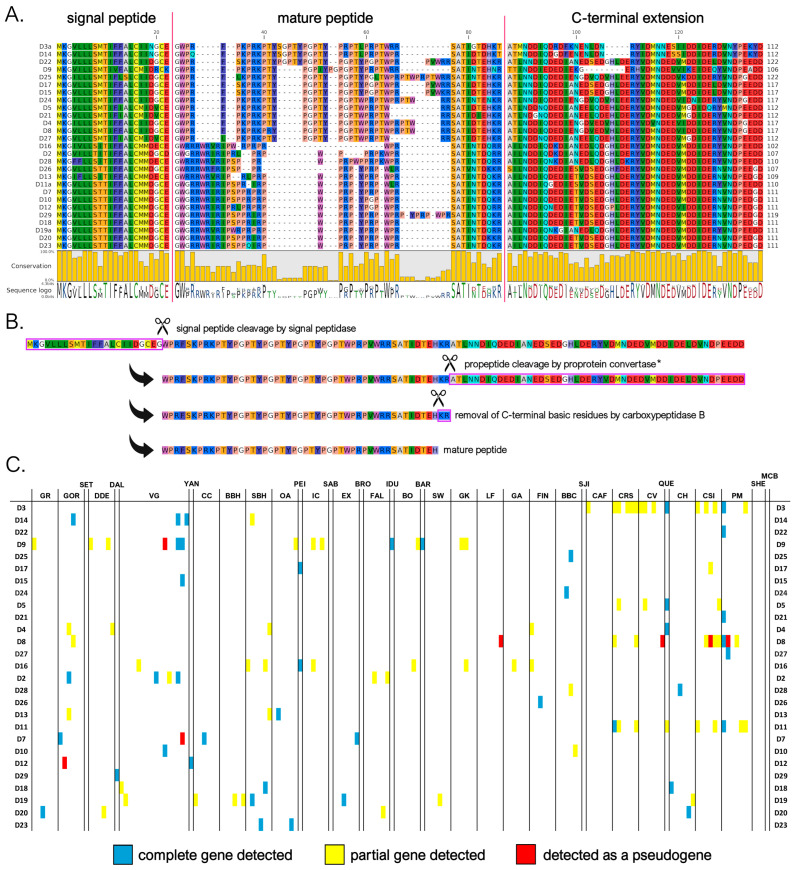
(**A**) Class D myticalins displayed in a multiple sequence alignment; (**B**) the events predicted to occur during the maturation of the precursor proteins, exemplified by myticalin D22; and (**C**) the occurrence of the sequences in sampling locations, where colors denote whether a complete (blue), partial (yellow), or pseudo (red) gene were detected. Please note that only a single representative precursor sequence is shown for each mature peptide. The complete multiple sequence alignment including all precursor sequences is available online on MyticalinDB. * Note the absence of a dibasic site in myticalin D3a, D9, and D14.

**Figure 6 genes-16-00816-f006:**
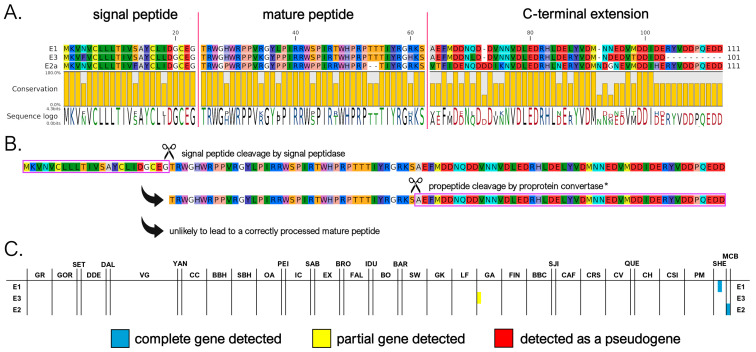
(**A**) Class E myticalins displayed in a multiple sequence alignment; (**B**) the events predicted to occur during the maturation of the precursor proteins, exemplified by myticalin E1; and (**C**) the occurrence of the sequences in sampling locations, where colors denote whether a complete (blue), partial (yellow), or pseudo (red) gene were detected. * Note the disruption of the canonical dibasic site in all myticalin E precursors.

**Figure 7 genes-16-00816-f007:**
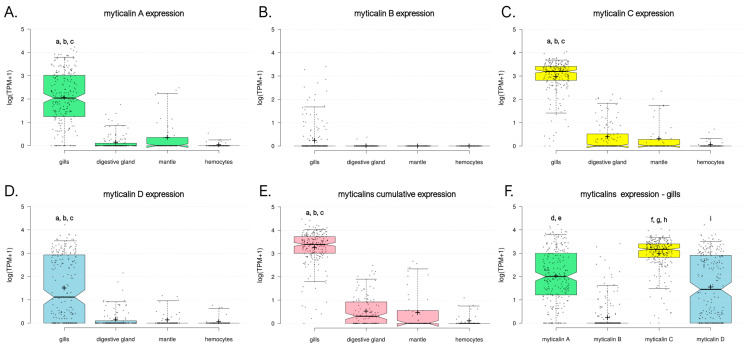
Expression level of myticalins, calculated based on the analysis of available RNA-seq datasets for species belonging to the *Mytilus* complex. Gene expression levels are reported as log10 (TPM+1). Each dot represents a single RNA-seq experiment and box plots indicate the median, 25th, and 75th percentile of expression ranges for each tissue. Whiskers indicate the 5th and 95th percentiles. The graphs report the cumulative gene expression levels of all myticalin sequences belonging to subfamily A (**A**), B (**B**), C (**C**), and D (**D**). No expression was detected for sequences belonging to subfamily E. (**E**) Shows the overall expression levels observed in the four tissues of all myticalin variants. (**F**) Displays a comparison between the gene expression levels of the four major myticalin subfamilies in the gills. Lowercase letters indicate statistically significant differences (i.e., *p*-value < 0.05), in the following pairwise comparisons among samples. a: gills vs. digestive gland; b: gills vs. mantle; c: gills vs. hemocytes; d: myticalin A vs. myticalin B; e: myticalin A vs. myticalin D; f: myticalin C vs. myticalin A; g: myticalin C vs. myticalin B; h: myticalin C vs. myticalin D; and i: myticalin D vs. myticalin B.

**Figure 8 genes-16-00816-f008:**
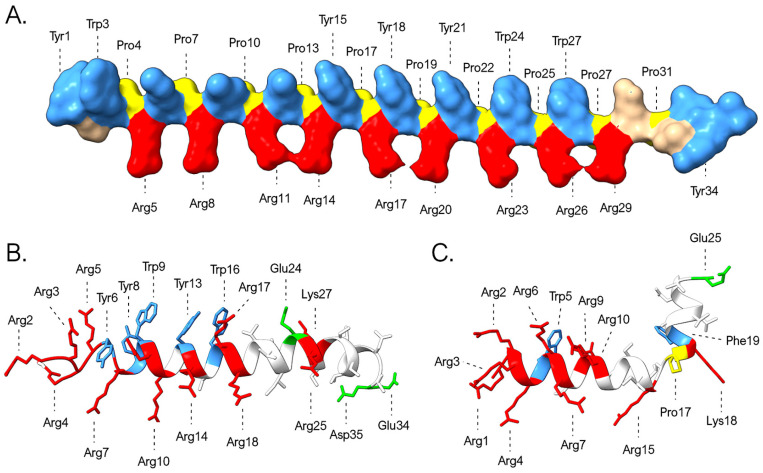
Three-dimensional structural models of myticalins A5 (**A**), C10 (**B**), and C7 (**C**) predicted by AlphaFold. The graph highlights the placement of prolines (yellow) and of the positively charged (Arg and Lys, red), negatively charged (Asp and Glu, green), and aromatic (Tyr, Trp, and Phe, blue) residues.

## Data Availability

All the peptide sequence data reported in this manuscript are available online on MyticalinDB: https://myticalindb.units.it/. The original data presented in the study are openly available in NCBI at the repositories/publications referenced in the Section 2 and in Appendix A.

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
