# Peer review of "Tracing the Evolutionary Expansion of a Hyperdiverse Antimicrobial Peptide Gene Family in *Mytilus* spp.: The MyticalinDB Resource"

_genes, 2025, doi:10.3390/genes16070816_

Round 1

Reviewer 1 Report

Comments and Suggestions for Authors

The entire paper is based on a prediction of mature peptide, which requires validation. 
Normally, when we have a GRKR-type motif, it is highly likely that cleavage takes place upstream of the basic site and glycine allows amidation of the peptide, as has been observed for antimicrobial peptides such as piscidin present in fish gills. 
It is essential to demonstrate or not the presence of the active mature peptide predicted by the authors, especially as the two papers referenced [21, 22] do not demonstrate this. The sequences indicated are certainly active, but the mature form of the peptide resulting from post-translational maturation, notably by convertases alone, has not been demonstrated. It is highly likely that carboxypeptidase and peptidyl amidating monoxygenase are involved. Purification and sequencing of the mature peptides and/or analysis by mass spectrometry would be required.

Author Response

The entire paper is based on a prediction of mature peptide, which requires validation.
Normally, when we have a GRKR-type motif, it is highly likely that cleavage takes place upstream of the basic site and glycine allows amidation of the peptide, as has been observed for antimicrobial peptides such as piscidin present in fish gills.
It is essential to demonstrate or not the presence of the active mature peptide predicted by the authors, especially as the two papers referenced [21, 22] do not demonstrate this. The sequences indicated are certainly active, but the mature form of the peptide resulting from post-translational maturation, notably by convertases alone, has not been demonstrated. It is highly likely that carboxypeptidase and peptidyl amidating monoxygenase are involved. Purification and sequencing of the mature peptides and/or analysis by mass spectrometry would be required.

We would like to thank the reviewer, since we appreciate these insights. This is certainly an important point which we have considered in detail and discussed previously (Leoni et al. 2017). Unfortunately, there are technical limitations that currently prevent the possibility of this analysis. Obtaining sufficient quantities of any single myticalin peptide for purification would require pooling a large number of individuals, since the net weight of the gill tissue of a single individual is very small. However, since myticalins are subject to extensive presence/absence variation, each mussel carries a unique combination of variants that sometimes, as we demonstrate in this paper, display subtle differences. Therefore, the purification of a single myticalin peptide would require the selection of pools of individuals characterized by the same genetic background (i.e. expressing the very same myticalin variants). Unfortunately, at present, large scale genotyping of mussels is not technically feasible, and no genetically uniform lineages of this non-model species are available for laboratory use.

Some alternatives could be explored, such as synthesising single peptides in eukaryotic production systems such as Pichia pastoris. However, we argue that this would introduce other problems, since many of the possible post translational modifications that would target myticalin in mussels would not be achieved in that production system. We hope that future developments allow this question to be explored more thoroughly, but at the moment, in this biological model, this is not possible. We have nonetheless tried to emphasise the uncertainty of post translational modifications, clarifying they’re likely to occur but  that we are unable to predict them e.g. in myticalins C and D which do not have the typical amidation site (section 3.5, considerations about the structure of myticalins; lines 638-645). We have also included a note in section 3.3 (lines 333-337) explaining why obtaining experimental evidence is currently unfeasible.

Reviewer 2 Report

Comments and Suggestions for Authors

The paper investigates gill-specific myticalins, which are cysteine-lacking antimicrobial peptides (AMPs) in mussels (Mytilus spp.), playing a role in mucosal immunity. The work is timely and relevant, particularly in the context of mucosal immunity in mollusks. However, several aspects of the manuscript could be improved for clarity, precision, and consistency. With minor revisions to language and clarity, and more specific detail in methods and results, the abstract will effectively highlight the strength and novelty of the work.

Results and Discussion: Given the lack of canonical cleavage motifs and expression data, it would be helpful if the authors more explicitly discuss alternative hypotheses for its origin (e.g., convergent evolution, horizontal gene transfer) or degenerate pseudogene status, possibly with support from comparative genomics or synteny analysis.

The manuscript would benefit from a more nuanced discussion on how PAV influences functional redundancy or diversity, perhaps with comparisons to other AMP families in mollusks or other taxa.

The reliance on bulk RNA-seq from heterogeneous gill samples limits the resolution of expression dynamics. Please elaborate on how this might affect interpretation, especially in the presence of highly similar paralogs. Could long-read transcriptomics or single-cell RNA-seq improve resolution in future studies?

The AlphaFold2 predictions are a highlight, particularly for myticalins A and C. However, these models should be interpreted cautiously. A more critical discussion of the limitations of these predictions—especially for small, disordered peptides—is warranted.

Figures:  Figures 2, 7 and 8 are hard to interpret without zooming. Including zoomed-in panels or providing downloadable high-resolution images could help.

Conclusions: Some parts of the text would benefit from improved wording, removal of redundancies, and a stronger connection to broader biological or applied contexts.

Author Response

The paper investigates gill-specific myticalins, which are cysteine-lacking antimicrobial peptides (AMPs) in mussels (Mytilus spp.), playing a role in mucosal immunity. The work is timely and relevant, particularly in the context of mucosal immunity in mollusks. However, several aspects of the manuscript could be improved for clarity, precision, and consistency. With minor revisions to language and clarity, and more specific detail in methods and results, the abstract will effectively highlight the strength and novelty of the work.

Results and Discussion: Given the lack of canonical cleavage motifs and expression data, it would be helpful if the authors more explicitly discuss alternative hypotheses for its origin (e.g., convergent evolution, horizontal gene transfer) or degenerate pseudogene status, possibly with support from comparative genomics or synteny analysis.

We thank the reviewer for this comment. We assume the reviewer is referring to the myticalin E subfamily which is the one missing the canonical cleavage motif and lacking expression.

We have expanded the discussion of the evolutionary origins of this subfamily (section 3.3.5; lines 528-533). Among the different hypotheses proposed by the reviewer, we believe the one that is best supported by the observed taxonomical distribution of the genes is if the five different subfamilies were present in the LCA of all Mytilus species, as paralogous genes. Over the nearly 80 million years of independent evolution of the Mytilus species complex and the M. californianus/coruscus lineage, we believe that some of the myticalin families were nearly entirely lost, or accumulated deleterious mutations in either of the two lineages. For example, myticalin subfamilies A and C are still present as functional genes in all species. However, the D subfamily is functional only in the Mytilus species complex, and is present as pseudogenes occasionally in the M. californianus/coruscus lineage, with the opposite occurring with the B subfamily (mostly present and functional in the M. californianus/coruscus lineage; discussed in section 3.3.2 - lines 415-421). Meanwhile, subfamily E probably lost its function in both lineages: we only observed a single E pseudogene in one out of 100 individuals in the Mytilus species complex, indicating the loss is ancient. Whereas the M. californianus/coruscus individuals all have at least one gene, although it's likely non-functional (given the missing typical cleavage site). There are several other species related to Mytilus (e.g. Crenomytilus, Trichomya, Gregariella) that may still retain functional myticalin E like genes, but at the moment no genomic data is available for them, so this would be an interesting future research avenue. The establishment of myticalinDB is our idea of a way to include additional sequences that are discovered, not just from the Mytilus genus.

Concerning the alternative hypotheses suggested for the evolution of the E subfamily: we don’t believe that convergent evolution is likely. The gene structure is extremely conserved compared with the other myticalins, and we have evidence that myticalin E genes are part of the same genomic cluster, suggesting they were originally tandemly duplicated paralogues. Horizontal gene transfer can be excluded for the same reason. To support this interpretation we have included  a supplementary figure (Figure S2) which reports the organisation of the genomic locus in chromosome level assemblies.

The manuscript would benefit from a more nuanced discussion on how PAV influences functional redundancy or diversity, perhaps with comparisons to other AMP families in mollusks or other taxa.

Thank you for this suggestion. Gene PAV is a concept that has been only very recently explored and accepted in animals and therefore just a few examples of hypervariable individual collections of bioactive peptides have been reported outside Mytilus, where multiple cases have been previously documented (e.g. myticins, mytilins, mytimycins, CRP-I, etc.). Nevertheless, we believe that interesting parallelisms can be drawn with the venom peptides produced by predatory snails, even though the mechanisms underlying this extraordinary molecular diversity have not been connected with gene PAV yet. While AMPs and venom peptides clearly play extremely different biological roles, they are both part of the so-called exogenome, i.e. the fraction of genes that encode protein products targeting exogenous targets, which are typically fast evolving and hyperdiversified. We have updated the text (Section 4; lines 659-682) to incorporate a more detailed discussion of the predicted functional consequences of PAV of AMP sequence diversity in light of these observations, underscoring the exogenome (i.e. the portion of the genome encoding products with exogenous targets) as a potential hotspot of adaptive innovation.

The reliance on bulk RNA-seq from heterogeneous gill samples limits the resolution of expression dynamics. Please elaborate on how this might affect interpretation, especially in the presence of highly similar paralogs. Could long-read transcriptomics or single-cell RNA-seq improve resolution in future studies?

This is an interesting observation - it is true that the presence of similar paralogs could have an impact on the accurate determination of expression levels of myticalin variants, and for this reason we chose to focus on cumulative estimates at the sub-family level. Due to the significant divergence of primary sequences among the five subfamilies, read crossmapping is very unlikely to affect these estimates. The future availability of isoseq data would definitely allow very accurate determination of the expression levels of the individual myticalin variants even in cases of extremely similar paralogs or allelic variants. On the other hand, single cell sequencing may allow us to clarify another key question about the expression of myticalins which is their association with specialised cells that may be present in the gill tissue. We have included a note on this in section 3.4; lines 596-600.

The AlphaFold2 predictions are a highlight, particularly for myticalins A and C. However, these models should be interpreted cautiously. A more critical discussion of the limitations of these predictions—especially for small, disordered peptides—is warranted. 

Once again we agree with this observation, for this reason we strictly relied on the predicted structure with the highest score (pLDDT >80) and consequently only discussed A and C without mention of B and D whose structure was far less reliable (discussed in lines 625-627). We included an extended discussion of this potential issue in the revised text by also considering the possibility that other post-translational modifications may occur and affect the folding of the peptide, or that these apparently disordered peptides may acquire an ordered 3D structure in hydrophobic environments, an occurrence documented for other invertebrate AMPs with strong compositional biases (lines 638-645). 

Figures:  Figures 2, 7 and 8 are hard to interpret without zooming. Including zoomed-in panels or providing downloadable high-resolution images could help.

We understand that the figures as viewed in the draft manuscript may be difficult to see in detail, however when published they should all be in full size and accessible to zoom in the online version.

Conclusions: Some parts of the text would benefit from improved wording, removal of redundancies, and a stronger connection to broader biological or applied contexts.

Thank you, we’ve revised the text, and made sure to simplify and consolidate some complex sections throughout. We have extensively re-worked the conclusion, and now include additional points regarding gene PAV, and connections to other biological organisms, specifically conopeptides found in cone snails.

Reviewer 3 Report

Comments and Suggestions for Authors

The study analysed the evolutionary expansion of an antimicrobial peptide gene family corresponding to several species included in the Mytilus spp. I think the study is well presented, justified, and discussed. Additionally, it includes a wide range of traditional and advanced analytical tools including genomic and proteomic ones. I think it can be accepted for publication. Before that, some minor aspects could be revised.

I would mention the following concrete aspects:

Abstract

Maybe it is somewhat long (over 260 words) for the journal requirements.

Keywords

Evolutionary expansion could be included.

Material and methods

No statistical analysis is presented. Was not it necessary ?

Results and discussion

I have the doubt whether the catching location of a single species or the maturation degree of individuals may have any effect on the myticalin qualitative and quantitative composition. If so, maybe some comments could be included.

Author Response

The study analysed the evolutionary expansion of an antimicrobial peptide gene family corresponding to several species included in the Mytilus spp. I think the study is well presented, justified, and discussed. Additionally, it includes a wide range of traditional and advanced analytical tools including genomic and proteomic ones. I think it can be accepted for publication. Before that, some minor aspects could be revised.

We thank the reviewer, and have endeavoured to address these minor concerns.

I would mention the following concrete aspects:

Abstract

Maybe it is somewhat long (over 260 words) for the journal requirements.

We have now reduced the abstract to 223 words.

Keywords - Evolutionary expansion could be included

This keyword has now been included.

Material and methods

No statistical analysis is presented. Was not it necessary ?

We appreciate this suggestion and reviewed our analyses, with particular reference to the analyses of gene expression levels. We have now included statistical tests comparing the differential expression levels of myticalins in different tissue types, in particular finding that expression was significantly higher in the gills. The additional method is described in lines 192-196, and in the results in section 3.4: lines 568-569, 572-573, 576-579, 590-591, and Figure 7.

Results and discussion

I have the doubt whether the catching location of a single species or the maturation degree of individuals may have any effect on the myticalin qualitative and quantitative composition. If so, maybe some comments could be included.

The reviewer is correct that the sampling location could have an effect on the collection of sequences we presented in the paper. While we used all available population data covering the vast majority of the mytilus species-complex range, we are nonetheless missing information for a few species from the southern hemisphere, and there is a strong discrepancy between the number of individuals represented in the Mytilus complex versus the M. coruscus/californianus lineage. As more genomic data becomes available, it will be interesting to observe if there are clear patterns in the geographic distribution of myticalins. This is a point we expand on in section 3.1 (lines 227-235).

We are indeed sure that the full collection of myticalins is much larger than what we present, and this is a point discussed in section 3.1. A primary motivation of the new online database is to oversee new additions as they are discovered in the future.

Regarding the question about the developmental stage of the individuals: this wouldn't have an effect on myticalin discovery since our focus was on the extraction and analysis of genomic DNA, which doesn’t change over the mussel life span. However, this factor would have had a strong impact if we were working on the transcriptome.

Round 2

Reviewer 1 Report

Comments and Suggestions for Authors

This paper presents a family of antimicrobial peptides, the myticalins, for which the mature forms have not been experimentally validated. A significant concern is the absence of purification and subsequent mass spectrometry analysis to definitively confirm the mature sequences produced and released by the mussel.

It is crucial to emphasize that the "mature peptides" depicted in Figures 2, 3, 4, 5, and 6 are, at this stage, putative sequences rather than experimentally validated mature peptides. Their actual mature forms require rigorous biochemical demonstration through techniques such as purification and mass spectrometry.

Questions Regarding Post-Translational Processing and Cleavage Sites:

The proposed mature sequences raise several questions, particularly concerning post-translational maturation and cleavage sites. Mature antimicrobial peptides are typically cationic, highly enriched in arginine and/or lysine, and often feature multiple domains with successive basic residues.

During post-translational maturation, cleavage events frequently occur at monobasic or dibasic sites. The definitive mature sequences are generally located between these sites and do not typically include them. This is especially relevant when a glycine residue precedes these cleavage sites, as is the case for Class A and Class B myticalins. For instance, if a GRKR site is present, it is highly probable that cleavage occurs after the glycine, potentially leading to amidation, a phenomenon well-documented in fish antimicrobial peptides.

  • Cleavage Site Designation: Given that the putative mature peptide sequences are highly cationic and rich in basic residues, why are the proposed cleavage sites not indicated before the basic sites? This is critical, as mature peptide sequences generally exclude these basic amino acids, which are often cleaved during processing.
  • Glycine and Amidation: The role of glycine residues preceding cleavage sites for potential amidation appears to be overlooked. As noted, the presence of a glycine immediately before such sites often facilitates an enzymatic reaction leading to amidation of the preceding amino acid. This mechanism is particularly pertinent for Class A and Class B myticalins, and the paper should address the possibility of amidation, especially in cases like a GRKR site, where cleavage after the glycine with subsequent amidation is highly likely, analogous to observations in fish AMPs.
  • Discrepancy in Cleavage Patterns (Class E Myticalin): The paper indicates a cleavage after a serine residue for Class E myticalin. This specific cleavage site raises further questions:
    • What is the rationale for cleavage occurring after serine in this instance?
    • Why do the dibasic sites observed in other myticalin sequences not undergo similar cleavage events?
    • Which enzymes are responsible for these types of cleavages, particularly the serine-specific cleavage? Have such enzymes been identified or characterized within mussel genomes?

Author Response

This paper presents a family of antimicrobial peptides, the myticalins, for which the mature forms have not been experimentally validated. A significant concern is the absence of purification and subsequent mass spectrometry analysis to definitively confirm the mature sequences produced and released by the mussel.

It is crucial to emphasize that the "mature peptides" depicted in Figures 2, 3, 4, 5, and 6 are, at this stage, putative sequences rather than experimentally validated mature peptides. Their actual mature forms require rigorous biochemical demonstration through techniques such as purification and mass spectrometry.

We understand the concerns of the reviewer and agree about the need to further emphasize the putative nature of the mature peptide sequences. The text has been amended to reflect these considerations and provide further clarifications to some of the specific queries posed by the reviewer, see lines 151-158, 213-216, 329-227.

Figures 2, 3, 4, 5 and 6 have been modified accordingly to include a more detailed overview of the hypothesized proteolytic cleavages and additional post-translational modifications the precursors of each of the five myticalin subfamilies may undergo. As we will explain in response to the queries below, we have identified a fundamental issue that went unnoticed in the previous round of review, since we realized that the structure of those figures (and in particular the presence of scissor symbols in certain positions) would have been misleading to readers unfamiliar with our two previous publications on myticalins.

We hope that these addition and the modifications made to the figures and their captions will help to clarify the maturation process, also solving some of the other potential issues that were highlighted below by the reviewer.

Questions Regarding Post-Translational Processing and Cleavage Sites:

The proposed mature sequences raise several questions, particularly concerning post-translational maturation and cleavage sites. Mature antimicrobial peptides are typically cationic, highly enriched in arginine and/or lysine, and often feature multiple domains with successive basic residues.

During post-translational maturation, cleavage events frequently occur at monobasic or dibasic sites. The definitive mature sequences are generally located between these sites and do not typically include them. This is especially relevant when a glycine residue precedes these cleavage sites, as is the case for Class A and Class B myticalins. For instance, if a GRKR site is present, it is highly probable that cleavage occurs after the glycine, potentially leading to amidation, a phenomenon well-documented in fish antimicrobial peptides.

Cleavage Site Designation: Given that the putative mature peptide sequences are highly cationic and rich in basic residues, why are the proposed cleavage sites not indicated before the basic sites? This is critical, as mature peptide sequences generally exclude these basic amino acids, which are often cleaved during processing.

Yes, we most definitely agree, and in fact the predicted mature sequences of myticalins, both in this manuscript and in our two previously published works, do not include KR (or GRKR) sites. All the sequences deposited in our online database https://myticalindb.units.it/ clearly report the sequence of the mature peptides without KR (or GRKR) sites, including C-terminal amidations whenever needed. As a matter of fact, all myticalin precursor sequences are include two different fields, i.e.:
1) “Mature peptide (AMP) before post-translational processing”

2) “Mature peptide (AMP) after post-translational processing”

We would also like to remark the fact that all the peptides lacking canonical cleavage sites (e.g. all myticalin E precursors) are associated in MyticalinDB with a warning flag “sequence is potentially not functional: missing canonical propeptide cleavage site”.

We believe that the cause of this misunderstanding lied in the organization of Figures 2, 3, 4, 5 and 6, which might have been misleading to readers unfamiliar with our two previous publications on myticalins, and we are grateful to the reviewer for identifying this issue so clearly.

As the reviewer noted, in all these figures the “mature peptide region” does indeed include such sites. However, we need to mention that our designation of the “mature peptide region” was not intended to represent the product of all the different types of subsequent post-translational modifications (e.g. the removal of C-terminal K and R by CPB and the amidation of glycine by PAM), since these could not be properly evidenced in a multiple sequence alignment (i.e. they occur in different positions in different myticalin precursors). On the other hand, the boundaries of the “mature peptide region” correspond to the two cleavage sites: on the N-terminal side, the cleavage is operated by signal peptidase, whereas on the C-terminal side it is performed by proprotein convertase. This is now clearly explained in the materials and methods section, lines 151-158, which also clarify the reason why these boundaries are shown in certain positions even in precursor proteins, such as myticalin E, which lack a dibasic site in a canonical position.

We have removed the scissor symbols in those positions to avoid any misunderstanding, and added to the updated versions of figure 2, 3, 4, 5 and 6 an additional panel to explain, step by step, the different cleavages and modification each of the five myticalin subfamily precursors are expected to undergo. In detail, the C-terminal KR or RKR sequence stretches resulting from the cleavage by proprotein convertase would be secondarily removed by the action of other enzymes, i.e. most likely carboxypeptidase B, and amidation would only occur at this stage for the peptides displaying a C-terminal free glycine residue.

Glycine and Amidation: The role of glycine residues preceding cleavage sites for potential amidation appears to be overlooked. As noted, the presence of a glycine immediately before such sites often facilitates an enzymatic reaction leading to amidation of the preceding amino acid. This mechanism is particularly pertinent for Class A and Class B myticalins, and the paper should address the possibility of amidation, especially in cases like a GRKR site, where cleavage after the glycine with subsequent amidation is highly likely, analogous to observations in fish AMPs.

As mentioned above, we have always considered all myticalin A and B mature peptides to be C-terminally amidated, as previously explained in Leoni et al. 2017, reported in MyticalinDB, and now better explained in the main text (see lines 362-365 and 405-408). Please also note that we have previously assessed the antimicrobial function of myticalins A and B in their amidated forms.

Discrepancy in Cleavage Patterns (Class E Myticalin): The paper indicates a cleavage after a serine residue for Class E myticalin. This specific cleavage site raises further questions:

What is the rationale for cleavage occurring after serine in this instance?

As mentioned in the main text, we do not believe that such cleavage occurs in that position, see L520-522 “This is further supported by the implied loss of proper maturation and, potentially, function. The peptides lack a canonical proprotein convertase recognition motif in the expected position, which is typically required for post-translational processing, even though a dibasic site potentially available for cleavage would still be present in a slightly offset position in myticalin E1 and E2 (Figure 6, panels A and B)”. As explained in the materials and methods section (lines 153-158), we define the C-terminal boundary of the “mature peptide region” thanks to an evolutionary criteria, i.e. based on multiple sequence alignment data.

Although we cannot definitively exclude the possibility of cleavage being carried out in an alternative position, all myticalin E sequences are most likely pseudogenes, as also suggested by the lack of expression. We have modified figure 6 to remove the scissors symbol, since we realized it was confusing, in particular in this case. The new panel B explains that we suspect that no cleavage occurs at all. Please also note that these sequences are marked as likely not functional in the online database.

Why do the dibasic sites observed in other myticalin sequences not undergo similar cleavage events?

We believe that the reviewer is referring to the occasional presence of other dibasic sites in non-canonical positions, such as -most notably- the arginine-rich myticalin C precursors. We have no definitive answer to this question, but we need to remark the fact that, even in other species, the cleavage or lack thereof of dibasic sites is context-dependent, meaning that flaking amino acids at the N- and C-terminal sides of the cleavage sites, as well as the 3D structure of the proteins may or may not leave a given dibasic site available for recognition and subsequent cleavage by proprotein convertase. Our prediction is based on indirect evidence dependent on the fact that the KR site marked in our figures is evolutionary well-conserved, despite being present in the context of a hypervariable precursor sequence, which is a strong indication of its functional relevance.

Which enzymes are responsible for these types of cleavages, particularly the serine-specific cleavage? Have such enzymes been identified or characterized within mussel genomes?

As explained above, we do not believe that myticalin E precursors undergo any proteolytic cleavage due to their likely pseudogenic nature. The same is true also for a few members of the other myticalin subfamilies, that we believe may represent the product of degenerated genes which lost their function.

In general terms, the main cleavage we expect to occur in all myticalin precursors after the conserved dibasic site may be due to proprotein convertases, which display a very broad taxonomic distribution. Nevertheless, to the best of our knowledge, none of these proteases have ever been functionally characterized in bivalves, and reports about their identification are extremely scant even in other closely related biological systems, such as cone snails (gastropod mollusks), where such modifications have been broadly documented to occur and target the precursors of bioactive peptides (e.g. conotoxins).

Reviewer 2 Report

Comments and Suggestions for Authors

The manuscript has significantly improved in quality. The authors have revised it in accordance with the reviewers' comments."

Round 3

Reviewer 1 Report

Comments and Suggestions for Authors

The new version is better, although the absence of additional MS/MS analysis is regrettable